# Unsure Theory: Ambivalence as Methodology

## Caitlin Merrett King

Glasgow School of Art, 167 Renfrew St., Glasgow G3 6RQ, UK; c.mking1@student.gsa.ac.uk

**Abstract:** Ambivalence is often regarded as a 'negative' emotion—an 'ugly feeling' as Sianne Ngai outlines—where not knowing and being unsure are seen as suspicious or mentally unhealthy. In this article, I outline the initial exploratory stage of the development of a new affective theory that I have termed 'Unsure Theory', in which ambivalence is observed as a mobile and aporetic state that, from an individual perspective, embraces the holding of multiple contradictory personal opinions. Unsure Theory also outlines ambivalence as an appropriate contemporary, meta-modernist response to late stage capitalism, our current socio-political moment, and its often negative impact on mental health. The aesthetics of ambivalence is explored through embracing a hesitant vernacular, an oscillating humorous, dry and ironic to sincere tone, and an internal, anecdotal first person voice that often addresses the reader. This exploration of Unsure Theory operates in an adjacent, feminist lineage of, and in homage to, Sad Girl Theory, as coined by writer, critic and artist Audrey Wollen, and Sick Woman Theory, by artist, writer and musician Johanna Hedva, as well as Lauren Fournier's critical responses to both. Written within the genre of art writing and in reference to my own interdisciplinary creative practice, this article exemplifies autotheoretical writing as an extension of contemporary visual art practice. This article is partially situated within my own personal experience of Cognitive Behavioural Therapy from 2020–2022 and reading the autotheoretical novel *Chelsea Girls* by Eileen Myles at the beginning of 2021. Through unpacking these personal experiences, I begin to outline an argument for embracing ambivalence, particularly within autotheoretical practice, where Unsure Theory seeks to repoliticise uncertainty towards a new generative, critical and personal perspective on not knowing.

**Keywords:** autotheory; memory; visual arts; art practise; contemporary art; social history; critical writing; art writing; ambivalence; affect theory

## 1. The Revolving Door

After a writing workshop in the Autumn of 2020, K suggests that I read *Chelsea Girls* (Myles 2016) because of Eileen Myles' rich descriptions of eating food.[1] I order the book online and several months later, at the beginning of February 2021, finally pull it out from the very ambitious and very dusty pile of yet-to-be-read books next to my bed. Sixty-five pages deep into the filthy opulence of *Chelsea Girls*, Myles kisses me with a reference to Robin, an Aquarius with whom they become infatuated, which, for a typical over-relating Aquarius such as myself, is as pungent as if they'd penned a love letter to me instead and hidden it there for me to find, on delicious page sixty-five: 'There she sat with her extraordinary stark white-face, a weirdly shaped skull, kind of cubist and long, with raven-ish black hair. I adored her because she was a masque. This combined with her sensibility, literary and scrupulous, made her essentially Aquarian to me, an endless revolving door' (Myles 2016, p. 65). Oh! To be described as a masque with a literary and scrupulous sensibility! The allure of being an endlessly revolving door! The Aquarius par excellence, it's me.

During a particularly painful portion of 2019, I had downloaded the astrology app Co-Star and would send my three most patient friends their horoscopes every day. My own became a comfort blanket that I could wrap myself up in and try to navigate my life

from. I would receive the pithy daily notifications every morning, often saving the good ones (i.e. mainly the ones that told me what I wanted to hear) for later, and study my compatibility with other friends on the app. I would read and re-read the sections that gave an overview of the broader focus of my life at that time, searching for something recognisable and relatable to make sense of what I was experiencing. Every day a new piece of advice to keep me going, a new comfort blanket that at times was incredibly helpful, and at other times completely overwhelmed my ability to formulate my own observations and corresponding emotional responses. To be completely honest, I was immensely sad and desperate and too sad and desperate to know how to really talk about or really feel how sad and desperate I was.

Also during this portion of 2019 I started going to a private therapist who specialised in Cognitive Behavioural Therapy, at the suggestion of a very good and sensible friend. After a few sessions I noticed that my therapist was repeatedly using the same word to describe my general behaviour and attitude towards my relationships: *ambivalent.* It felt like a pretty crushing insult at first and was framed to me as an unhealthy coping mechanism, part of also having an insecure attachment style. I bought *Attached: Are you Anxious, Avoidant or Secure? How the science of adult attachment can help you find—-and keep—-love* (Heller and Levine 2010), at my therapist's suggestion and read it immediately, learning about the complexities and legitimacies of different attachment needs, trying to unlearn my ambivalence, seeing it as a total failure. I completed all the sheets my therapist gave me, listened to the podcasts they recommended and started to actually feel better. I went nearly every week, then two weeks then monthly, for three years until last week when I decided that I was ready to stop.

An important thing to mention here is that my plan is not to recount any of my therapy notes to you or really any particularly personal details because this essay, this hypothesis about the importance of embracing ambivalence, is not really about me, even though it definitely is all about me. This statement offers a pretty accurate rubric for autotheoretical practice in my opinion, where the author writes from their own perspective often in relation to personal experience in order to frame a new theoretical framework.[2] In this sense, autotheoretical writing is about writing *from* the self. Within this exploration of the possibilities that Unsure Theory offers, I hold my own experiences close and write from the place of processing them. Unsure Theory comes *from* me.

It was Myles on delicious page sixty-five of *Chelsea Girls* who made me realise that it was possible to think twice about this framing of ambivalence as a bad thing. The 'ambivalent dance', another CBT term, that I completely relate to and see myself performing often, the fleetness of foot, is Myles' endless revolving door illustrated. The revolving door spins, the ambivalent person rotates between positions, Myles' glamorous ambivalent whirls through my mind, raven-black hair flying behind her, her tongue both measured and observant, and critical and withholding behind her masque. (Myles 2016, p. 65) Don't get me wrong, I completely understand the possible negative outcomes of this lack of commitment to one position or mode—the anxiety of being positionless and position-full—-but what I want to argue for here is quite simple: that to have mixed feelings about something or someone, and not to be sure about how you feel is inherent in the nature of being a human. It is especially pertinent to the experience of living in the twenty-first century under late stage capitalism and its inherent complications, specifically its impact on mental health. Unsure Theory, in this way, begins to make a claim for ambivalence as both intrinsic to contemporary individual experience, but also necessary in unpicking the lack of nuance and misapprehension of mixed feelings that is present within hegemonic thought and upheld by late stage capitalism.[3] Ambivalence, rather than being suspicious or mentally unhealthy, can be a very useful creative, aesthetic, political, critical, and personal tool.

Here, in this article, you might notice my hesitant voice, my first person, often anxious and uncertain language with moments of hedging and skirting around the point sometimes. This informal register is used here just to exemplify an attempt at embracing a language

of uncertainty, to talk about ambivalence with an ambivalence. It might also not be consistently used which again illustrates a genuine commitment to a lack of commitment. I will continue to discuss Eileen Myles' *Chelsea Girls* later but my lines of thought will go elsewhere in a kind of stream of encounter. The trouble with this is that you, the reader, might lose interest or lose track of my thinking. Despite my anxiety at this, I want to hold open this tacit risk of writing from one's own personal experience, especially when its intentions are not necessarily to present a clear and unwavering argument. And so, quite appropriately, this might result in you, the reader, like me, feeling pretty ambivalent too, and maybe this, as a success of the relatability of the feeling of ambivalence, is not really a bad thing.

Some context on my creative practice: I studied Fine Art at undergrad but after a few years moved away from making physical objects, and began programming exhibitions and events in Glasgow, sometimes collaborating with others and performing with others as well as trying to do some writing. Because of this I see my writing practice as an extension of my art practice or rather that writing sits inside my art practice like a grainy blue egg in a plump nest. This confusing and often messy place is often academically referred as 'art writing', as per the MLitt Art Writing that I am currently enrolled on at the Glasgow School of Art, which encourages interdisciplinary 'creative, philosophical, critical and theoretical approaches to writing about, writing with, and writing as art' (Haynes 2017). Art writing is in 'the situation of the fulcrum', as previous programme leaders of the now defunct MFA Art Writing at Goldsmith's University of London, Maria Fusco, Yve Lomax, Michael Newman and Adrien Rifkin define in '11 Statements Around Art Writing' (2011). Paradoxically, they continue, art writing seeks to avoid definition and is rather a selection of possibilities: 'Art Writing is re-invented in each instance of Art Writing, determining its own criteria' (Fusco et al. 2011). The slippery trickiness of art writing makes way for the slippery trickiness of feeling which makes way for the slippery trickiness of ambivalence.

More specifically, I write about contemporary art and culture, and exhibitions that I see in Glasgow mainly, and I see these sites as being a perfect place for me to experiment with and enact an ambivalent language. Sometimes you go to an exhibition and you do not really know what to think, do not have an opinion or just are not sure what the artist is trying to do and that is ok, you know? Because not everything can be for everyone, right? Especially art or writing about art or writing about writing about art, which are all so inherently subjective. So, to use autotheory as a vehicle for my own art writing practice is about acknowledging these gaps between both individual and collective experiences. I utilise the first person and my own experiences to build a theoretical perspective, not to enforce my own singular one, but to encourage inconclusive, experimental thought that rebukes the societal need for neoliberal, hegemonic thought. Returning to Fusco, Lomax, Newman and Rifman: 'Art Writing names an approach within contemporary culture that, in wanting new potentials, embraces writing as a problematization of the object of art, its dissemination and forms of exhibition' (Fusco et al. 2011).

## 2. Against Defining Ambivalence

What is important to admit at this point is that I am not completely certain of my own definition of 'ambivalence'. My definition comes from my own lived experience of ambivalence and is therefore shaky and unreliable. Subjectivity allows space for definition but definition does not often allow space for true subjectivity—a statement which is also true for when an absolute definition of autotheory is attempted. Unlike people, definition cannot ever contain multitudes—'Do I contradict myself?/Very well then I contradict myself/(I am large, I contain multitudes.)'—Walt Whitman provides a perfect stanza-long un-definition of the multitudinous ambivalent (Whitman 1891–1892). Currently, I think my definition actually straddles something between ambiguity and ambivalence which feels pretty appropriate to my whole project really. I log into the Oxford English Dictionary online to look for something to push up against.

According to the OED 'ambivalence' is: 'The condition of having contradictory or mixed feelings, attitudes, or urges regarding a person or thing' (OED Online 2022a). It is: 'the condition of being undecided about a viewpoint or course of action, or of being unconvinced by the merit of something; the state or fact of being contradictory or inconsistent' (OED Online 2022a). And it is also, in psychological terms: 'The coexistence in one person of profoundly opposing emotions, beliefs, attitudes, or urges (such as love and hate, or attraction and repulsion) towards a person or thing' (OED Online 2022a). A third and final definition of ambivalence listed in the OED relates to 'a vibrating body': 'the fact or state of absorbing sound according to its acoustic properties as an emitter. Only in *the principle of ambivalence*' (OED Online 2022a). A vibrating body that takes in what it gives out, which sounds more like a principle of equitable empathy than an ambivalent one to me. Within my thinking about how ambivalence could be initiated as a positive personal tool, where mixed feelings are acknowledged and held rather than suppressed, however, empathy is certainly a key component, and tacitly consequential to a true ambivalence that holds mixed feelings and contradictions, born from the self and absorbed from and in (and out of) time with others.

Scrolling further down the page I notice that the OED online contains the records of the 'obsolete' definitions as well. Interestingly there are none listed for 'ambivalence' but for 'ambiguous' there are several: 'Originally and chiefly of language: having different possible meanings; open to more than one interpretation'; 'Of an event, course of action, etc.: that has an uncertain outcome or conclusion; that could unfold in a number of different ways'; 'Not reliable; inconsistent.' (OED Online 2022b). It feels especially relevant, within my project and towards a loose definition of ambivalence as it sits within Unsure Theory, to take on all the old and no-longer-relevant definitions. I seek a loose definition because if I was looking to pin down a finite one then I would not be maintaining my own ambivalent aims of ambivalence, would I? And this refusal to be definitive might bring me closer to an idea of ambivalence as multitudinous, mixed and inconsistent: a truly ambivalent (and ambiguous) ambivalence.

## 3. Poeta Che mi Guidi

Back to the start of 2021 and back to *Chelsea Girls.* In an appropriate autotheoretical effort, I follow the Maggie Nelson approach to writing an essay that I heard her describe recently in a Zoom lecture at Glasgow School of Art. When beginning to write, she will go back through each book pertinent to the writing to find the notes/underlinings/page corners that she has turned over on previous reads. She will then compile said notes and write to plug the gaps between. And so, I flick through *Chelsea Girls* scouring the pages for studious marks. Back to page sixty-five which is coincidentally the only page corner that I've turned over and the page that contains the only words that I've underlined, which are again: 'This combined with her sensibility, literary and scrupulous, made her essentially Aquarian to me, an endless revolving door' (Myles 2016, p. 65). I will leave that there because really I've said most of what I would like to say about the revolving door above—my adopted visual cipher for a methodology of ambivalence.

The only other thing, and most central part of the story to mention, is that Myles' protagonist (Myles) and Robin (the ambivalent Aquarius) have a sad and tumultuous on-and-off love affair. The final words of this short story within *Chelsea Girls* ends, 'Now I sit in this incredible silence. I do not know why.' (Myles 2016, p. 70) They are honest, unsure and vulnerable with the reader, yet they do not over-explain their emotions in an attempt to keep you interested, but rather, courageously risk an ambivalent reception by laying their feelings bare and open-ended right at the close.

I flick back through to Myles' introduction to look for some of their grounding and more signs of ambivalence. As that's the first thing I read in the book, it's probably a good place to start again. This new introduction was written in 2015 when the book was rereleased by Harper Collins during a significant boom of interest in autotheoretical writing, 2015 also being the year that Maggie Nelson's *The Argonauts* was published (Nelson 2015).

And so this introduction is Myles reflecting on their writing eleven years after the initial publishing in 1994 which feels like a pretty long time really, especially seeing as it took them thirteen years to write it. What they mainly want to tell us in the introduction, they say, is that, in *Chelsea Girls*, they needed to say what they thought was 'real'. (Myles 2016, p. xi) Because as they say on the next page, referring to themself in the third person and with their 2015 pronoun: 'how would she ever be real unless she told the story of it'? (Myles 2016, p. xii). Indeed, I want to be 'real' and for my first person voice to be 'real', and how will I ever be 'real' unless I talk about it?—another autotheoretical rubric for you there.

It's like in CBT where past experiences are used to make sense of present and ongoing behavioural patterns and emotions—my autotheoretical 'real' is a processing of my present into my past through writing. This impulse to be 'real', through the conveyance of my own emotions, is central to my autotheoretical critical mode and central to processing those very emotions. It's like in CBT when they ask you to write a letter to yourself from the perspective of a friend; fictioning, and specifically autofiction—as we might suspect partially of *Chelsea Girls*—is a useful distancing tool in order to gain new and therapeutic perspectives.[4] There is a tension though—as displayed in my writing here with not wanting to tell you too much about myself, not wanting to make too much of an example of myself, because I reserve the right to keep some things to myself, you know? My own autotheoretical voice (I do not really want to speak for anyone else here or at all) exercises the right to be multiple voices—casual, anecdotal and personal (auto), and academic, conceptual (theoretical) in tone—and many different selves—real and/or not real and/or partially real.

To be 'real', Myles lives and writes and cries with the grief that everything will be gone eventually, including themself. 'Who was she' (Myles 2016, p. xii). This is not a question though, there is no question mark, is there? And Myles is not going to give me a single answer either. 'Life is shocking to me, is and always was, full like a garden of so many selves' (Myles 2016, p. xii). I take each story in this book to be a different self of Myles, the book is a kind of polyphonous garden within a garden. A garden full of different selves that all make up their own self, but not completely as I am sure that there are many other selves to be found in many other different books and other different gardens, already written and not yet written. 'Writing was', they say, 'the only space I could go to show how utterly real it all was' (Myles 2016, p. xiii).

'*Chelsea Girls* initially and finally is a lot of things it's not' (Myles 2016, p. xii). Which is really the perfect finite-not-finite ambivalent statement on form—Thank you, Eileen! It is not the film that they actually wanted to make, it is long form poetry and a novel and myth, and 'accounts of lots of recuperated disasters' (Myles 2016, p. xii). The recognition and inclusion of failure also feels important within the project of ambivalence. I do not want to use the word confessional because of how gendered it has become, but maybe it's time to reclaim it? Myles states in an interview with *Rookie Mag* on whether they relate to their work being called 'confessional' that, 'only in the sense that I'm an ex-Catholic! I did like confession. I like getting things off my chest in some way.' (Benjamin 2015) Their first-person narrativising of trauma ('recuperated disasters`') is a legitimising of experience that also feels to me like a kind of self-therapising. Indeed, 'Who was she'. And so I also state (partially and indirectly) about myself, with a confessional-non-confessional ambiguity, 'who was she (no question mark).'

Another recommendation from K in early 2021 is this Ellena Savage essay that I keep coming back to. I redownload the PDF and scroll through looking for the bit where she talks about how the origin of the personal essay was historically masculine, despite it the being the domain of 'the margins', i.e., women/people of colour/nonbinary people/queer folk, since the mid-twentieth century (Savage 2017, p. 3). It was also, and is still contemporarily, towards 'the liberal democratic project of self-fashioning'. Savage deems this kind of essay to be a 'subject position-oriented personal essay'. An arm of the autotheoretical, the personal essay creates a direct line between author and reader, and becomes an 'object of mediation', thereby holding a tacit complication, an ideological problem of attempting

to be both individual and collective, and therefore (sometimes) creating or replicating hegemonic thought (Savage 2017, p. 2). And so, Savage asks, how emancipatory can the personal essay be if part of what it's doing is just reinforcing a neoliberal sense of individual experience? To avoid this, I believe, the idea of the individual voice *could* (I will avoid any imperative language here to stay with the ambivalent voice; again, I do not want to speak for anyone else but myself here) be complicated through the embracing and enacting of the ambivalent voice, which itself is composed of a complicated, inconvenient, emotional and unfinished set of voices that seek to refuse singularity or definition through their very existence. I google through the avalanche of Eileen Myles interviews from 2015. Endless, endless interviews conducted by a wave of ambivalent (mainly) millennials gushing over *Chelsea Girls*, although Myles seems the most unfettered. In one interview in *The White Review,* Myles declares, 'Art has to finally be the place where something gives and the form is forever different' (Dimitrova 2016).

Back to the *Rookie Mag* interview (very appropriate mid-2010s feminist territory) where Tova Benjamin talks about the infamous (and only) filmed interview with Clarice Lispector where she recounts how a male professor told her that he could not understand *The Passion According to G.H.* whereas a seventeen year old girl told her that she loved it and could not stop re-reading it. Here, Lispector makes the point that understanding is not about intelligence but about feeling. Myles responds saying, 'It's so weird because someone did just ask me in an interview if I wanted to be understood and I said no, I want to be felt' (Benjamin 2015). Through feeling, searching for feeling, holding feeling and in turn being felt by others, is all part of this complicating of the individual voice, where the individual is acknowledged as feeling multiple. Myles continues, 'people always think that means, "emo," and it's like no, you're a sentient being, you're alive' (Benjamin 2015). *Real.*

Back again to the beginning. Whilst reading *Chelsea Girls* at the beginning of 2021, I google Eileen Myles and find a joyous image of them sprawled out, arms behind their head. They have a tattoo on their upper arm, an Italian script that at first glance through the thick pixels I read as 'porta che mi guidi', translating as 'a door to guide me'. I lose my shit, it's the Aquarius, their revolving door! However, in actual fact, on closer inspection, I can just about make out that it reads 'poeta che mi guidi'. Another google pings up the extended quote:

> Io cominciai: "Poeta che mi guidi,
>
> guarda la mia virtù s'ell' è possente,
>
> prima ch'a l'alto passo tu mi fidi.
>
> English translation:
>
> And I began: "Poet, who guidest me,
>
> Regard my manhood, if it be sufficient,
>
> Ere to the arduous pass thou dost confide me.
>
> (Longfellow 1867)

A further google tells me that the quote is taken from 'Canto II' at the beginning of *Inferno,* where Dante as protagonist is guided by the ancient Roman poet Virgil on a journey through the underworld. Virgil finds Dante wandering astray in a wood, tormented by a she-wolf, a lion and a leopard, three wild beasts that represent the sins of wantonness, violence and malice. Dante follows Virgil, cautiously at first, through the gates of Hell ('Lasciate ogne speranza, voi ch'intrate'—'Abandon all hope ye who enter') in 'Canto III' where, on the shores of the river Acharon, the Uncommitted reside, those who lived their lives without taking sides and with concern only for themselves.

Here, I sit on the banks of the Acharon, an *Uncommitted*, refusing to side with one position or another, caring about myself (but not exclusively). Virgil is the poet who guides Dante and I suppose quite literally up to this point, Myles is the poet who has whirled me around and guided me. In *The White Review* interview, Myles offers a further reading of the tattoo, 'It's when Dante met Virgil and he basically wanted to know if he would be able to

walk through hell, and so he said, "Poet, take my measure" you know, "I think I can do it"'(Dimitrova 2016).

## 4. Mixed Ugly Feelings

I develop a nervous tick for picking at my eyebrows causing the skin to become dry and flaky. I then pick further to get rid of the dry skin, standing very close to the bathroom mirror at work rubbing each brow furiously in rapid succession. I try to remember to moisturise in the morning but just cannot fight the temptation to pull out the thick hairs. I leave a hot trail of skin flakes and wiry hairs wherever I go. Tiny little snowflakes collect in the inner hinge of *Ugly Feelings* (2005) by Sianne Ngai, between pages 2 and 3.

Moving away from ambivalence being regarded as a 'negative' emotion, where 'not knowing' and being unsure are seen as suspicious or mentally unhealthy, I shuffle ambivalence up alongside Sianne Ngai's outline of other negative affective states. In *Ugly Feelings*, Ngai's aim is to analyse the negative affects that she sees as '*a*moral and *non*cathartic' (Ngai 2005, p. 6). The ugly feeling of ambivalence that I am offering into this mix is less a feeling and more of a brimming bucket that holds all of the 'negative' ugly feelings—ugly feelings being mostly the result of conflicted or mixed feelings, uncomfortable products of a state of turbulent ambivalence. This brimming bucket of ambivalence, I believe, can be defined as both *a*moral and moral, and *non*cathartic and cathartic.

I have to read and re-read the introduction to get a good grasp of what Ngai is talking about. Three years later, I read and re-read it again before starting to write this paragraph. Brain fizzing from eating too many sweets in the library, I drag my blunt pencil across the thin creamy page underlining the phrase, 'I prefer not to' (Ngai 2005, p. 2). Ngai introduces me to the perfect ambivalent right from the top of *Ugly Feelings:* Bartleby, the inexpressive protagonist in Herman Melville's short story (Melville 2016), 'Bartleby, the Scrivener: A Story of Wall Street'. The world of Bartleby accurately exemplifies the flat affect generated and nourished by late stage Western capitalism. When Bartleby repeatedly refuses to work, using the emotionally ambiguous phrase, 'I prefer not to', Ngai questions his political intention. Is he a radical, revolting against the choke hold of repetitive labour as generated by capitalism? Or is he just idle, passively passive, privileged even in his ability to say, 'I'd rather not do that' to every single task his employers on Wall Street set him, and further engaging in the individualistic neoliberal attitude that partaking in capitalism has imbued within him? His passivity, at least, I believe is an active choice (maybe there's no such thing as a passive passivity anyway?), and whilst Melville's aims also are unclear, Bartleby's refusal can certainly be framed as a passivity towards simply being a cog in the capitalist machine. He clogs up the machine until he is forcibly removed and later dies impoverished in jail.

For Ngai, Bartleby's world is also the ideal example of how literary space—and I would broaden this to any creative space—is the ideal site to investigate ugly feelings and their potential for 'critical productivity' (Ngai 2005, p. 2). She describes art's self-consciousness of being (or at least just feeling like its being) left behind due to its often unempirical aims within an empirical, commodified society, where art (and literature) is a cordoned-off bourgeois and powerless thing. It is this superfluity that renders the aesthetic domain of art and literature capable of a vast creative flexibility and, in turn, a consummate space to examine the politically ambiguous work of negative emotions. I think I can just about get my head around all this and it makes me think about Lauren Berlant's outlining of intimacy as a space that problematises the Victorian imposition of the public and private dichotomy (Berlant 1998, p. 283). The bourgeoisie created these public discursive spaces, such as salons, cafes, and exhibition spaces (my addition), to be able to perform their private, critical thoughts to each other. Coming back to Ngai: 'the very act of thinking the aesthetic and political together [ . . . ] is a prime occasion for ugly feelings' (Ngai 2005, p. 3).

I scratch my eyebrow again as I turn the page, scattering a short snow fall on page 4. Ngai pulls me back and warns me against over-romanticising, quoting Paolo Virno (Virno 1996): in the workplace, 'capitalism's classic affects of disaffection [...] are neatly absorbed

into the wage system and reconfigured into professional ideals' (Ngai 2005, p. 4). My insecurities and anxieties about work, that are generated from the very act of being at work, result in a new desire to be adaptable and flexible, bending to the restrictions of neoliberal capitalist drive—-my anxiety is fed back to me *ad infinitum* and I gulp it down. I do want to be less cynical about this though, although cynicism (is cynicism an ugly feeling?) also feels somewhat necessary within my project of ambivalence really. Ngai points out that our emotions are not too clearly defined anymore when it comes to contemporary political and social action. That we are not just angry or fearful about the current shambolic state of politics, we are now also less classically passionate about politics, feeling the petty and less glamorous emotions of anxiety, passivity, insecurity and *ambivalence*.

Scrolling towards a place of something that may come, I arrive at Craig's lecture and watch it on my phone flipped sideways leaning against my cup of coffee (Pollard 2020). Craig explains to me, over a photo of Tony Blair cheersing a pint of lager with Jacques Chirac, that we have now arrived at a *metamodernist* moment. I open a new tab on my phone and scroll through the 'Notes on Metamodernism' website: 'However, despite, or rather *because* of this, a yearning for meaning—for sincere and constructive progression and expression—has come to shape today's dominant cultural mode' (Turner 2015). Our metamodernist era explodes us into an oscillating state of possibility, taking lessons from both postmodernism and modernism, embracing apathy and affect, an active yearning for utopias: 'Each time the metamodern enthusiasm swings toward fanaticism, gravity pulls it back toward irony; the moment its irony sways toward apathy, gravity pulls it back toward enthusiasm' (Turner 2015).

So then what else should I be feeling if not ambivalent right now, clasped between sincerity and irony, like a depressed and confused Subway sandwich (I am sprucing up the brimming bucket analogy to a newer and juicier, more contemporary consumer capitalism one) exploding with all the possible options, cheeses, meats, tomatoes, olives, gherkins, sweetcorns, onions, a million relishes, salad cream, mayonnaise, all oozing and splurging out the sides?

Scrolling back to C's lecture, and then back to Ngai again, an interview in *The White Review* where Ngai talks about the affective conditions of ambivalence, about how ambivalence is an 'appropriate response to late stage capitalism' (Brazil 2020). Ngai's ambivalent reading calls for a recognition of the inherent ambivalence held within capitalist and commodity form aesthetics: universal and compromised. She remarks on how our affective mixed and conflicted responses are tied up in our observation of the ethical and moral lack in late stage capitalism. What should we *do* with this innate metamodernist ambivalence then? It is this ambivalence and the ensuing ambivalent critical reading that makes us ripe for the 'critical productivity' that Ngai champions (Ngai 2005, p. 2). I return to flaky page 2 of *Ugly Feelings* and give each eyebrow another firm scratch.

A few weeks ago, I explained what I am trying to write about in this article to D, an art therapist friend. D points out that maybe, like with mental illness, ambivalence is not a contemporary condition symptomatic of our current socio-political climate, but that it is only just now that we have developed the language to really be able to articulate how we actually feel or at least to attempt it or even just to say, 'I'm unsure'.

## 5. Unsure Theory

This paper is an initial explorative attempt towards redefining and repoliticising ambivalence as a generative, critical and personal perspective, that I have coined 'Unsure Theory'. This new theoretical framework is built upon the shaky foundations of ambivalence that I have laid out above and in homage to Audrey Wollen's Sad Girl Theory[5] and Johanna Hedva's Sick Woman Theory,[6] as well as Lauren Fournier's criticisms of both (Fournier 2018). Unsure Theory, like Sad Girl Theory and Sick Woman Theory, is based around an exemplifying and centering of lived experience but maintains focus on the mixed resultant affective responses. It's all about centering affect—that is how you *feel* about it *all*—and embracing and expressing those mixed ugly feelings, through critical and creative

forms, such as interdisciplinary art writing and the autotheoretical. As an open ground for experimentation, autotheory can be used as a site to explore how ambivalence can be championed within everyday life towards an acceptance of mixed feelings and expanded nuanced thought, against a neoliberal individualistic approach. This outlining of Unsure Theory does not offer many suggestions as to how it can actually be applied but rather seeks to draw together some conceptual frameworks in order to explore my own understanding of ambivalence as a critical attitude or temperament. Unsure Theory brims like the Subway sandwich. Unsure Theory is unsure of itself even, I suspect.

I find myself stuck in a corner of the internet from the first half of the 2010s when *Chelsea Girls* was re-published by Serpent's Tail in the UK, and Audrey Wollen's Sad Girl Theory was at peak hype. I scroll through Wollen's Instagram feed but there is no hint of Sad Girl Theory and its archive @tragicqueen has been deleted too. Maybe Wollen only thought it was a theory for that era, maybe she did not want to be immortalised like that on Instagram anymore? I google and find an interview with her from 2014 on *i-D:* 'I want to stand with the girls who are miserable, who do not love their body, who cry on the bus on the way to work. I believe those girls have the power to cause real upheaval, to really change things' (Barron 2014). In Sad Girl Theory, Wollen repeats the selfie aesthetics of Instagram back to itself in knowing critique, representing and objectifying her own image in order to dismantle oppression. I am not too sure about this and think like Audre Lorde, who disagrees about the potential for the master's tools to be used successfully in taking down the master's house: 'they will never enable us to bring about *actual* change' (Lorde 2018, p. 19).[7]

Sad Girl Theory is also an attempt, through the posting of selfies photoshopped into historical artworks of women, to re-historicise and re-politicise sadness and to offer an alternative to 'hyper-positive' contemporary feminism. I click through Sara Ahmed's *Feminist Killjoy* blog, back to 2014, looking for a glimpse of feminist zeitgeist as cultural context and stumble upon the entry, 'Selfcare as Warfare'. Of course, 2014, the selfcare era! Ahmed's blog is the ultimate archive of anti-neoliberal queer feminist refusal theory, especially the refusal of the societal (and gendered) imperative to be happy; Ahmed is the realest Sad Girl. She insists that only true emancipation from patriarchal oppression—specifically for disabled, queer, female and racialised people—can be achieved through 'the ordinary, everyday and often painstaking work of looking after ourselves' (Ahmed 2014), and maybe not engaging with the tools of oppression, e.g. Instagram. I thought about making an Instagram account to sit beside this article but then I realised the depth of the hole that I might be digging myself into. I could always delete it I suppose. And that allowing-myself-to-change-my-mind would feel quite ambivalently relevant, wouldn't it? So here is a non-committal and fairly anonymous attempt at least for now anyway (Merrett King 2022).

I am worried that Unsure Theory is enforcing the Sad Girl Theory position of a sad, white, able-bodied, middle-class, privileged woman using theory as cultural social capital, as Lauren Fournier criticises (Fournier 2018, p. 650). I mean, maybe that is what I am doing already? Or is that what I will actually be doing when I inevitably post about this on my own personal Instagram account once it's published? Or am I just promoting myself and that is fine? Or is that what Fournier calls 'self-branding'? (Fournier 2018, p. 653)[8] I think there's something very millennial about the idea of ambivalence as social capital so I may as well own that. Deeply knotted into an ambivalent attitude is a penchant for apologia but I am trying to shake that whilst also checking my privilege, which feels like a lot of hard work to be honest. And so, to build on Wollen's feminist framework of rehistoricising affect, whilst rooting firmly in mixed feelings (including sadness), Unsure Theory drips towards the desire to repoliticise uncertainty and not knowing. Unsure Theory also drips towards an excessive language—of tone, of superlatives, of hedging phrases, of honesty—that welcomes further nuance of the experience of affective response and its engagements. As Lauren Elkin writes, 'A little less self-censoring; a little more Conviction, please.' (Elkin 2013, p. 157). I am trying, Lauren, really I am. As Fournier states, 'autotheory has the

potential to resist the ossification of selfie-care and instead stand as a contemporary practice of theorizing that is accessible beyond the borders of academic institutions' (Fournier 2018, p. 650). Here I am, a complicit and confused student, in this article, doing meta-autotheory within an academic setting, but mainly on the subject of existing outside of it really. But anyway, ambivalence is nothing if not totally against an ossification of everything, always pendulous, always moving. Unsure Theory moves: it is 'the vibrating body' (OED Online 2022a). I scroll through images, I am overwhelmed, I am overjoyed, I repeat.

I scroll through *Topical Cream*, in search of Johanna Hedva's 'Sick Woman Theory' first published in 2016. Hedva speaks about the frustration of being chronically ill during the Black Lives Matter protests in 2014, and again (in this updated version) during Russia's continued terror on Ukraine in early 2022, bed-bound, unable to participate in physical protest on the streets: 'I share links and doomscroll, then need to log off, fatigued and weeping as the images of war, oppression, and genocide proliferate on my screen' (Hedva 2022a). The doomscroll is fatigue-inducing hyper-engagement, both public and not-public and not-real-public. In agreement with Ahmed, Hedva proclaims, however, that the most anti-capitalist thing we can do is: 'To protect each other, to enact and practise a community of support. A radical kinship, an interdependent sociality, a politics of care' (Hedva 2022a).[9] It's hard not to be hard on ourselves but these are neoliberalism's rules of maximisation of profit at whatever mental or physical cost, and neoliberalism will not look after us. We must look after each other and ourselves, against a neoliberal individualism, in a collective therapy. As Fournier observes, Hedva's theory, along with Wollen's, involves, 'a practice of "failing" to be as active as one would like to be—as a feminist activist, as a scholar, and so on' (Fournier 2018, p. 648). To return to the algorithm, online, this failure, plus Ahmed's idea of refusal, surfaces as the *glitch*. Legacy Russell outlines how the glitch breaks away from the gendered physical body and is 'a call to action as we work towards fantastic failure': failure to be commodified, failure to be capitalised, failure to be defined (Russell 2020, p. 3).

Like Sick Woman Theory, Unsure Theory is inherently a practice of failure. It is also a practice of rewarding failure that productively complicates binaried 'positive' and 'negative' affective responses. As Halberstam outlines, despite the negative affect that accompanies failure, 'it also provides the opportunity to use these negative affects to poke holes in the toxic positivity of contemporary life' that demands our compliance with societal norms and the idea that, in life, there must be winners and losers—the cornerstone of capitalist logic (Halberstam 2011, p. 3). Halberstam continues to say that as well as failure being a way of being in the world, it is also a mode of 'unbecoming [that] propose[s] a different relation to knowledge' (Halberstam 2011, p. 23). Unsure Theory seeks to reconfigure a hegemonic understanding of knowledge by championing uncertainty, not-knowing and the importance of changing one's mind.

Unsure Theory also fails in its attempt to be defined, as that is against the very nature of being unsure. It fails itself too: it is anti-capitalist yet complicit; at odds with the academy yet crouched here in this Special Issue. I thought about writing a manifesto in homage to Johanna Hedva's Sick Woman Theory, but who is the ambivalent person-subject? It's sometimes me, yes, at least here it is, but then why would I write a manifesto just for myself? I am not here to make any rules, and Unsure Theory is not about being dogmatic. And then, what are the aesthetics of the ambivalent? It's what I am doing here, I suppose. This paper is an attempt at exercising some of the aesthetics for sure, but will also fail to be complete, because that is not the point, is it?

Unsure Theory lives and breeds rampantly online. I doomscroll passionately through Instagram. I grew up as the internet grew up in the 1990s and so its codes are woven deep into the fibre of my being. Scrolling now, on my phone, which I've picked up without realising, on Instagram, I feel an *algorithmic ambivalence*—it's the digital revolving door. It's Ngai's capitalist gimmick: 'it is a form we marvel at and distrust, admire and disdain, whose affective intensity for us increases precisely because of this ambivalence' (Ngai 2017). It is this ambivalent response to the algorithm of Instagram—that gives me what I like

and what I might like without even asking for it but based on what I directly tell it that I like—that intensifies my affective response and my dedication to the object of the phone (Ngai 2020, p. 23).[10] I scroll through images, I am overwhelmed, I am overjoyed, I repeat.

I scroll through images, I am overwhelmed, I am overjoyed, I repeat. I find a link to a podcast on *Polyester Zine*'s Instagram about Goblin Mode, the pandemic-induced slob craze to reveal the 'worst' parts of yourself online (See Gamble and Young (2022); Paul (2022)). But, as Iona Gamble and Eden Young reflect, 'is this just another bid for influencers and privileged women to perform relatability online?' (Gamble and Young 2022) Unsure Theory does not seek relatability. It is only that probably everyone feels unsure at some point and even then I am not sure about that. Maybe some people never feel unsure. Is that what it looks like to have a secure attachment style? (Heller and Levine 2010). Unsure Theory is sometimes in semi-Goblin Mode, and it's friends with the Sick Girl and the Sad Girl too.

I put down my phone and then immediately pick it back up again, scroll and find a meme about 'dead face' (Castro 2022a). Two white female characters from the TV series *Euphoria* stare blankly alongside another white female-looking person that I do not recognise. A hacked screenshot of a *Mailonline* article (Greep 2022) now reads, 'Move over duck pout, it's all about DEAD face! Influencers are adopting a "dissociative" gaze with rolled eyes and a blank expression - because it lets them express *their reaction after engaging with any of the decrepit, zombified abominations that pass for contemporary art and literature'*— italicised to highlight the modification to the original article title. Interestingly, the original *Mailonline* title states that dead face is about demonstrating 'beauty ironically'; the meme offers ambivalence as stand in for negative criticality, felt so strongly that it is post-verbal or anti-verbal, whilst also offering an ironic criticism of the *Mailonline*'s policing of young female-identifying people (Greep 2022). Also, interestingly (and somewhat relevantly for my practice) the meme was made by Jordan Castro, a New York-based author who recently wrote *The Novelist: a Novel* (Castro 2022b), an meta-autotheoretical piece about a man who tries and fails to write an autobiographical novel, distracted from his goal by social media, the banality of daily chores and overwhelming internal thoughts. So then, towards the 'dissociative' gaze with rolled eyes as expression of my reaction after engaging with some decrepit, zombified abominations that pass for contemporary art and literature? Not really, but kind of. Unsure Theory in practice could look a bit like that, I suppose.

After noting the blurring of 'criticality, accessibility, and complicity (with capitalism, with neoliberalism, with what Sharma calls "selfie-care")', Fournier concludes 'Sick Women, Sad Girls, and Selfie Theory: Autotheory as Contemporary Feminist Practice': 'I end my discussion of these fourth-wave practices on this ambivalent note, as I perceive an ambivalence at the heart of these autotheoretical projects themselves' (Fournier 2018, p. 656). There is ambivalence in this attempt to do it all, to have my cake and eat it too, to be critical, accessible and complicit with capitalism, then critical of that complicity for an audience (on Instagram) in order to present something that attempts an accessibility or seeks to be widely relatable. But then also against anything championing hegemonic thought and seeking categorisation, towards a non-neoliberal, personal but not individual, unsure polyphonic choir of hedgy voices, or it could be a non-verbal, meta-modernist, overwhelmed, multitudinous blank stare into the abyss: that might be Unsure Theory.

## 6. Afterword

I make a note-to-self to ask my therapist what they think of the state of my ambivalent attitude now that we're finishing our sessions together after three years. It would be good to see some improvement, I suppose, to get a rating on a scale of one to ten, from then and from now, something quantifiable to prove to myself that I am 'better' or at least 'better' than I used to be. I realise that this feels counter-intuitive; an affective response is not quantifiable and probably should not be capitalised on, however, but I still think I will ask out of interest. After our final session, I realise that I've completely forgotten to ask them what they think of the progression of my ambivalence. It's ok. It's not the point really, is it?

**Funding:** This research received no external funding.

**Institutional Review Board Statement:** Not applicable.

**Informed Consent Statement:** Not applicable.

**Data Availability Statement:** Not applicable.

**Acknowledgments:** All my thanks to the editors of this Special Issue Cat Auburn, and Katherine Baxter for their generous guidance and encouragement; to Elizabeth K. Reeder, the Essaying class, and Laura Haynes for reading the initial versions of this manuscript; to Craig Pollard for his inspiring research that was vital to this writing; to Grace Denton for her boundless enthusiasm; to David Roeder for wise words; to Kate Taylor and Kate Morgan for sharp recommendations and ardent reading; and my greatest thanks of all go to my therapist.

**Conflicts of Interest:** The author declares no conflict of interest.

## Notes

[1]   *Chelsea Girls* is an autobiographical novel written by poet Eileen Myles. Their writing is autotheoretical in nature in that they use real life experiences to challenge dominant Western heteronormative thought through the foregrounding of the complexities of queer, feminist experience. Myles states in the preface for the 2016 edition of *Chelsea Girls,* that for them, writing was the only space where they could truly show their true form; 'I think to be female and strange and to want art so much and be drunk and high even waking up from that, all of it and really to have lived' (Myles 2016, p. xiii).

[2]   Lauren Fournier outlines autotheory as, 'a term to describe the practices of engaging with theory, life, and art from the perspective of one's lived experiences; an emergent term, it is very much in the zeitgeist of contemporary feminist and queer feminist cultural production today' (Fournier 2018, p. 641). She also contextualises her research into autotheory as being based in what Amelia Jones refers to as 'self-imaging' (Jones 2006) and offers that autotheoretical practice, as well as interdisciplinary visual arts practices, 'bears consideration within the scholarly spaces of autobiography studies' (Fournier 2018, p. 642).

[3]   Fournier concludes 'Sick Women, Sad Girls and Selfie Theory: Autotheory as Contemporary Feminist Practice' by stating the political capabilities of autotheoretical work: 'As a transmedial, transdisciplinary, and transnational practice, autotheory has the capacity to trouble dominant epistemologies and approaches to philosophizing and theorizing, exposing the problematics of maintaining conceptual separations between self and theory' (Fournier 2018, p. 657).

[4]   Myles states in the preface that they are 'trying to figure out if *Chelsea Girls* is a novel or a memoir or a collection of stories (or whether it's really even a book at all)', which is suggestive of the fact that the text contains both fictional and non-fictional elements. It also points to the potential for fluidity and interdisciplinarity of medium that is inherent to autotheoretical practice (Myles 2016, p. xi).

[5]   Sad Girl Theory is a research project and theoretical framework devised by writer, critic and artist Audrey Wollen on her now defunct Instagram account @tragicqueen. The project outlined how, 'female sadness and self-loathing is not a singular experience to be ashamed of, but actually a form of empowerment that can ultimately unite women' (Tunnicliffe 2015).

[6]   'Sick Woman Theory' is an essay written by Johanna Hedva that was first published on the now defunct *Mask Magazine* in 2016. It was republished in *Topical Cream* with the accompanying text, 'Why It's Taking So Long', a further commentary on the difficulty of maintaining artistic practice specifically when disabled and living in a world designed for able bodies, and during the COVID-19 pandemic (Hedva 2022b).

[7]   Emphasis my own. Ironically, the American photography artist Richard Prince used images of Wollen that she had posted on @tragicqueen without her permission as part of his 'New Portraits' series first exhibited at The Gagosian in 2014; see (Barron 2014).

[8]   'Indeed, given the history of feminist performances of the self in art, one wonders what it is that Sad Girl Theory brings to feminism other than what appears to be a savvy move at self-branding in light of the influence that framing one's work as theory has for young women with class mobility who have been exposed to theory in art school' (Fournier 2018, p. 653).

[9]   'Because, once we are all ill and confined to the bed, sharing our stories of therapies and comforts, forming support groups, bearing witness to each other's tales of trauma, prioritizing the care and love of our sick, pained, expensive, sensitive, fantastic bodies, and there is no one left to go to work, perhaps then, finally, capitalism will screech to its much needed, long-overdue, and motherfucking glorious halt' (Hedva 2022a).

[10]  See further discussion about the affective promise of objects in (Berlant 2011) and (Ahmed 2010).

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
