# Peer review of "Unsure Theory: Ambivalence as Methodology"

_arts, 2022_

Round 1

Reviewer 1 Report

Overall, this article offers an interesting reflection on the ways in which ambivalence might be considered a productive way to think through one’s own autotheoretical practice. However, while it claims to outline a new theory which the author calls ‘Unsure Theory’, this is too big a claim for the scope of the article. In fact, in the abstract, the author states ambivalence is ‘re-garded as a teachable aporetic state where multiple contradictory opinions can be held towards a more equitable, ethical critical mode. Unsure Theory also outlines ambivalence as an appropriate contemporary, meta-modernist response to late-stage capitalism and the pressures of neoliberal individualism’ (p.1). These claims are not really met in the article (how might it be teachable? What do you mean by ‘a more equitable, ethical mode’?).

Instead, the article offers a laudable first exploratory phase of developing such a theory. I would advise the author makes this clearer from the start, presenting it as such rather than the presentation of a complete new theory. The article mainly sums up a range of different points which could be related to the idea of this theory, rather than making an overarching case for Unsure Theory. Its strength also precisely lies in its development of a present-tense style which guides the reader along and makes them see the literary and theoretical encounters made first-hand (i.e.: ‘Scrolling towards a place of something that may come, I arrive at C’s lecture and watch it on my phone flipped sideways leaning against my cup of coffee.’) There are some great observations in this article and the writing is overall pleasurable to follow. Addressing the reader is an effective way of engaging with the reader, but the more colloquial insertions (‘I promise’ / ‘you know’ / ‘would I?’) I found a little grating and I think the piece would be stronger if these were edited out.

I list here below some suggestions for revision and some things to think about in the revising of the article:

- Adjust the aims and scope of the article in the abstract to more correctly mirror the achievements in the body of the article.

- The references to ‘late capitalism’ need unpacking a little more – they’re often thrown in, but not fully engaged with. Can you unpack what you understand this to mean and how its effects are felt? 

- On p.2 you write:

‘Please note here that I’m not interested in recounting any of my therapy notes to you or really any details of my personal anguishes. I’m not going to go into any detail really because this essay, this hypothesis about the importance of embracing ambivalence, isn't really about me, even though it definitely is all about me – a statement which actually offers a pretty accurate rubric for autotheoretical practice in my opinion.’ 

 Could this statement be unpacked and thereby nuanced more? It also comes across slightly aggressively/defensively. And it’s clear that your experiences with therapy are a starting point for the essay, so it feels a little glib to be so dismissive here. I think this is, in a way, an interesting response which could perhaps be reflected upon – it namely ties in with accusations often levelled at autotheoretical texts/art which condemn and dismiss it as ‘navel-gazing’ behaviour. How do you see the relationship played out here (and in the theory you propose) regarding subjectivity/collectivity, relationship between self and world / …? A thought: is your hypothesis ‘all about me’; or does it start from ‘me’? To me, it seems this is an important difference.

- On p.3:  ‘It [the informal register] might also not be consistently used which again illustrates a genuine commitment to a lack of commitment.’

 is ambivalence equivalent to a lack of commitment?

Then you go on: ‘I’m going to talk more about Eileen Myles' Chelsea Girls, I think, but it might not be very rigorous and lines of thought will probably go elsewhere in a kind of stream of encounter. The trouble with this, like a lot of autotheoretical practice, is that you, the reader, might lose interest or lose track of my thinking, but really I think that’s ok, because you probably don’t know me personally so it’s quite a big ask to ask that you stay with me and my wavering the whole time. I’m not trying to be ungenerous, I promise, I just don’t think that life, or writing about life or writing about art (my interest as I describe next) is always going to be one hundred percent interesting and therefore this might result in you, the reader, feeling pretty ambivalent too, and maybe this isn’t really a bad thing.’ (p.3)

 This section similarly feels slightly defensive (see above); I recommend changing the tone here and again, delving into what anxieties are being played out here. What do you mean when you talk of ‘a lot of autotheoretical practice’? Are you thinking of particular pieces/artists/.. or are you paraphrasing a particular kind of critique of this work here? I understand that the point here is allowing space for ambivalent affects on the part of the reader, but it’s not as successful as it could be due to this more defensive tone (you appear to invite dismissal, almost).

- Section 2 on ‘Against defining ambivalence’: whilst I understand the need to not define ‘ambivalence’ too strictly, your discussion of Walt Whitman’s phrase highlighted the fact that this article currently contains a major contradiction (it is not about me / it is about me, subjectivity / collectivity). Could this be elaborated upon?

- The OED reference to schizophrenia is mentioned but not elaborated upon – what is its significance?

- You acknowledge the out-of-datedness of the Notes on Metamodernism website – and yet implicitly continue to find it a useful way of describing a particular era (i.e. it’s also used to describe this era in the abstract). Can you elaborate on this? Why/what parts of this theory appear/are useful and worth keeping a hold of?

- ‘…although there may be correlations between being female and ambivalent, my actual main concern is building a theory around only my own experience, rather than speak to a whole gender––is this self-mythologising? I make some claims and invite criticism. I am anti-expert. I am only myself.’ 

This could be considered a little careless in that it perhaps suggests the claims made by Myles and Wollen are guilty of speaking on behalf of a whole gender yet they are making important feminist claims about the ways in which women are perceived and treated – could this be rephrased, elaborated upon, nuanced? The claim ‘I am anti-expert’ also seems very laden in our current political context in which ‘expert’ seems to have become a tainted word….

Perhaps do away with the final line ‘I just guess I’ll never know.’?

Then, finally, some minor things to be amended:

‘Also during this portion of 2019 I started going to a private therapist, specifically Cognitive Behavioural Therapy, …’ (p.2) ïƒ  ‘specialised in CBT?’/rephrase?

This confusing and often messy place,…’ (p.2-3) ïƒ  sentence is lacking a main verb.

first person narrativising of’ (p.5) ïƒ  ‘first-person narrativising’

p.6: double-check the Italian translation – as far as I was aware, the Italian ‘virtù’ should be translated as ‘virtue’ rather than ‘manhood’?

p. 10 Lauren Elkins should be Lauren Elkin.

Author Response

Many thanks for your encouraging and helpful comments and edits. I am really grateful to you for engaging so thoroughly with the article. I will address each of the points from your review report in bullet form. 

  • I have updated my abstract as per your suggestions that I am working 'towards' a theory, in the first stages of exploration as opposed to presenting a final theoretical framework.
  • I have edited out some of the colloquialisms that you mentioned were grating at times, some of these remain for effect where I am sitting with uncertainty.
  • I have tightened up my references to capitalism; at times I had conflated capitalism with 'hegemonic thought' or 'neoliberalism'. At times I have used it as a contemporary short hand and like you say, not completely unpacked the actual complications of capitalism that I am referring to.
  • In regards to your notes on page 2 and 3 I have loosened my phrasing slightly to make the tone seem less defensive or anticipatory of criticism. I have kept some moments in to stay in touch with the anxiety of making my claims for ambivalence. 
  • I am ok with the contradiction within the writing
  • I have deleted the references to schizophrenia and the notes of metamodernism website being out of date as they are not completely relevant and take the writing off on a distracting tangent.
  • I have acknowledged your comment about the feminist claims of Wollen, Hedva and Myles' use of gendered terminology and rephrased my own reasoning, as well as deleting the word 'expert'– as you say it feels too loaded here to engage with. 
  • I have deleted the final line as per your suggestion
  • I have amended the minor grammatical corrections that you suggested. 

Reviewer 2 Report

In my opinion, the text submitted by the Author may be regarded (almost solely) as a literary work of art – an inspiring, original essay, abundant with vivid metaphors, (e.g.  the image of revolving door appropriated from Maggie Nelson’s Chelsea Girls). The admittedly picturesque language makes the reading very pleasant; however, some phrases are too colloquial. (The sentence: „I lose my sh.. !” needs to be removed; p. 6). One one hand, analysed in purely aesthetic terms, the text is highly valuable; yet, on the other hand, it does not fully meet the criteria of a scientific article. The structure of the text is clearly laid out and justified (by the method itself). Some interesting and significant references have been made to academic/scientific research, (ranging from Eugen Bleuler’s  classic psychiatry, to Sianne Ngai’s Ugly Feelings). Nevertheless, some references and remarks are inadequate in terms of scientific/ theoretical inquiry, (listed below).  Therefore, I recommend publishing the paper in Arts journal, but, at the same time, I strongly suggest inserting it in Essays section.

Let me enumerate some remarks/questions/suggestions below in bullet points. They do NOT need to be addressed by the Author, if the text is published in Essays section.

- References made to non-scientific „theories”, like e.g. Sad Girl Theory (Audrey Wollen), or Sick Woman Theory (Johanna Hedva), where such one-day „theories” are taken as serious, are acceptable only in literary texts;

- Inappropriate, in scientific terms, is to point at „the astrology app Co-Star” (1);

- The Author quotes Eileen Savage’s (historiographic) notes on writing an essay (2017). I think, Emily LaBarge’s Essay as Art Form of 2016 would provide no less interesting food for thought;

- The words „capitalist” /”capitalism” were used 14 times and, in my opinion: too easily combined with the notion of „thought hegemony” (used 4 times). I find no convincing proof to this linkage. (And – even as being very critical to capitalist encroachments – I find no convincing proof for that in scholarly texts. After all,  isn’t Melville’s Bartleby an INDIVIDUAL who refuses to COPY other authors? Economic individualism is too easily and too often confused with individualism as understood in strictly philosophical terms);

- I love the idea of the „Unsure Theory” proposed by the Author. „Errare humanum est”, and indeed, we (re)create ourselves (or: our multiple Selves) while wandering, through intellectual journeys.  In my (singular!) personal view uncertainty is opposite to totalitarian thinking. However, „championing not-knowing” (p. 8), both as scienfific attitude and social practice, would have catastrophic consequences. (I do not even dare to think of applying it to medical sciences);

- I do not think that writing about art is (inevitably and only) a subjective matter. I would rather define it as relational, and – if academic or critic – always at least in some part objective. But, the Author has the right to their own point of view.

Author Response

Many thanks for your comments and edits, and for taking the time to offer such generous and thorough feedback. In relation to your concerns about the positioning of my article within this special issue, I have discussed your review with the the editors who have advised that the methodology and style of my paper is appropriate to this special issue. 

I have taken on your comments about my use of  the word 'capitalism' and have amended several mentions. Thank you also for the suggestion of Emily LaBarge's thesis on the essay as art form.

Reviewer 3 Report

I congratulate the author on submitting a somewhat 'risky' paper that advocates for the value of ambivalence and discloses personal information to support the argument. The concept of 'Unsure Theory' is original and has merit in counteracting binary thinking predicated on a single fixed positionality. As outlined in the paper, the idea also resonates with a metamodernist sensibility of "oscillation", capturing the difficult socio-political moment in which we find ourselves.  

While the voice (personal, confessional/conversational and anti-intellectual) aligns with the theoretical position, the structure of the article could benefit from a more traditional approach to scholarship at times. For example, it would be valuable to introduce and contextualise key texts (e.g. Chelsea Girls), theoretical ideas (e.g. Sad Girl Theory) and terms for the reader (e.g. autotheoretical practice) when they are mentioned. There is also scope to expand connections between 'Unsure Theory' and allied theories such as Wollen's 'Sad Girl Theory'. Providing a more systematic discussion of these theoretical pre-cursors would help make an argument for the significance of a new model built on ambivalence.

There is also capacity to edit the document to ensure a logical flow of information (building to key insights) and to avoid over-complicating discussion with unnecessary details (i.e. information that is not directly relevant or connected to the central argument or later discussion points such as the astrology paragraph in the introduction). 

At present, the paper provides good insight into the development of the Unsure Theory framework. However, its potential for resistance or how the concepts can be applied (and to what end) are yet to be convincingly articulated. Consider more explicitly outlining how "...it (ambivalence) can also be a very useful creative, aesthetic, political, critical, and personal tool." What does it offer?

The concept of 'Unsure Theory' is interesting and has strong potential. With some further attention to structure and content in order to strengthen the central argument, the paper would be suitable for publication.  

Author Response

Many thanks for your helpful comments and enthusiasm. As you suggested, I have added in further explanation of the key terms, texts and theoretical concepts that I have mentioned, by introducing each in the endnotes when they are initially mentioned in the paper. I have also expanded connections between Unsure Theory and Sad Girl Theory and Sick Woman Theory, offering further comparison at times; I have also restructured this section so that my unpacking of SGT and SWT open the section, giving the reader more contextual framing before I discuss Unsure Theory further.

I have decided to keep in the beginning section where I discuss astrology as this is relevant to my unpacking of Myles' Aquarius in Chelsea Girls and also strengthens my first person, personal tone exemplifying a meta-autotheoretical approach.

I have also outlined that this paper on Unsure Theory is an initial exploration attempt towards redefining and repoliticising ambivalence as a generative, critical and personal perspective that draws together some conceptual frameworks in order to explore my own understanding of ambivalence as a critical attitude rather than offering many suggestions as to how it can be applied.